# Health outcomes in hospitalised and non-hospitalised individuals after COVID-19, an observational, cross-sectional study

Malin Nygren-Bonnier [1,2] ✉, Anna Svensson-Raskh[1,2], Linda Holmström[2,3,4], Anna Törnberg [1,2], Annie Svensson[1,2], Daniel Loewenstein [5,6], Malin Regardt[2,7], Mikael Björnson[8], Carl Hallberg[1,2], Mike Kemani[2,3], Anita Mc Allister[9], Joakim Körner Gustafsson[2,9], Alexandra Halvarsson[1,2], Urban Ekman[2,10], Linda Nordstrand[2,7], Susanne Guidetti [2,7], Lena Anmyr[2,11], Maria Bragesjö[12,13], Jenny Åström Reitan [2,3], Farzaneh Badinlou[12,13], Oili Dahl[14,15], Eva Åkerman[14,15], Pär Villner[16], Petter Brodin[4,17], Kenneth Caidahl [5,6], Marcus Ståhlberg[18,19], Artur Fedorowski [18,19], Magnus Sköld[18,20], Michael Runold[18,20], Judith Bruchfeld [8,21,22] & Elisabeth Rydwik[1,2,22]

## Abstract

**Background** Both hospitalised (H) and non-hospitalised (NH) individuals may have different symptoms and impairments after COVID-19. We aimed to explore symptoms, mental and physical health after initial COVID-19 for both groups of individuals and the association between physical and mental impairments in relation to self-rated health status and to identify different cluster profiles.

**Methods** Participants were recruited between June 2020 until December 2022 at the Karolinska University Hospital, Sweden. Data was collected at first assessment after COVID-19 and consisted of demographics, medical history, symptoms and results from physical function tests and self-reported questionnaires.

**Results** Here we show that among 931 participants, the H-group are older (mean age 56.7 years) and predominantly male (72%), while the NH-group are younger (mean age 44.4 years) and mostly female (84%). Fatigue, dyspnoea, joint pain, paraesthesia, and chest pressure are common symptoms reported across all participants. Physical function is lower than predicted in both groups and the NH-group have higher prevalence of depression and fatigue. These impairments together with dyspnoea, number of symptoms and sick leave are also associated with reduced self-rated health. Four specific cluster profiles have been identified, and 66.4% of the participants have severe to moderate impairments.

**Conclusions** Regardless of the initial level of care approximately two-thirds of the participants exhibit various physical and mental impairments associated to self-rated health after COVID-19. We propose that defining specific cluster profiles is crucial for tailoring management of post-COVID sequelae. Further long-term studies are needed to understand recovery trajectories to optimise targeted interventions.

## Plain language summary

This study examined symptom burden and physical and mental impairments in individuals with post–COVID-19 condition (PCC), comparing those who had been hospitalised with those who had not. Data from 931 participants (July 2020–December 2022) included medical history, physical tests, and questionnaires. Fatigue, breathlessness, joint pain, tingling, and chest pressure were common. Both groups showed reduced physical function, while depression and fatigue were more prevalent among non-hospitalised individuals. Four cluster profiles were identified, with most participants reporting moderate to severe impairments affecting self-rated health. These findings underscore the importance of comprehensive assessment and long-term follow-up to support recovery and guide rehabilitation strategies for people living with PCC.

Impairments in physical function, mental health, cognition and health-related quality of life after COVID-19 are reported in several studies[1–8]. In Sweden, over 2.8 million cases of COVID-19 infection have been micro-biologically confirmed, with over 111,000 people requiring hospitalisation due to COVID-19 since the beginning of the pandemic; worldwide, there are over 775 million confirmed cases of COVID-19[9]. The term 'post–COVID–19 condition' (PCC) is used to describe residual or late symptoms persisting after COVID-19 SARS-CoV-2 infection[10]. Depending on the study population, 7–35% of cases with earlier SARS-CoV-2 variants report residual symptoms and long-term disabilities[3,7,11,12]. Both hospitalised

and non-hospitalised individuals may, after their initial case of COVID-19, develop chronic symptoms in different organs and systems[3,13]. Common presentations in patients with PCC include fatigue, exercise intolerance, dyspnoea, headache, autonomic dysfunction, neurocognitive conditions and increased risk of stress, depression and sleep disorders[3,5]. The diversity of symptoms in PCC also impacts activities of daily living[4,6,14].

For the vast majority, COVID-19 causes relatively mild symptoms during the acute phase. However, as experienced at the beginning of the pandemic, initial disease can also present as a more serious condition, including respiratory failure requiring intensive care, which was commonly observed during the first and second COVID-19 waves[15]. Patients who have been cared for in an intensive care unit are at increased risk of physical and mental residual conditions, the so-called post-intensive-care syndrome (PICS), which can overlap with symptoms and conditions occurring in PCC[16]. However, individuals with an initial case of COVID-19, which did not require hospitalisation, also developed persistent symptoms and functional impairment[3,6]. Other risk factors for PCC have been shown to include female sex, old age and comorbidities[17]. Thus, initial disease severity and level of care (hospitalised or non-hospitalised) are not always predictive of the complexity of residual symptoms and impairments in PCC. Previous studies[5] are mainly based on self-reported data and not on objectively valid measurements. Five years after the pandemic outbreak, there is still a lack of high-resolution data regarding mental and physical recovery from PCC and whether PCC differs between non-hospitalised and hospitalised individuals. Therefore, the overall aim of this study is to explore demographics, symptoms and disabilities after initial COVID-19 illness among both hospitalised and non-hospitalised individuals, to analyse the associations between physical and mental impairments in relation to self-rated health, and to identify specific cluster profiles. Our findings reveal a high prevalence of physical and mental impairments across both groups, which are associated with reduced self-rated health. The identification of four distinct cluster profiles underscores the complexity and heterogeneity of post-COVID sequelae, offering new perspectives for targeted clinical assessment and rehabilitation.

## Methods

### Study design, setting and study population

This cross-sectional study is part of the observational longitudinal project ReCOV: recovery and rehabilitation during and after COVID-19 and is a collaboration between several departments at the Karolinska University Hospital and Karolinska Institutet. In this study, baseline data from clinical assessments are presented, and the full study protocol and methods are described in detail elsewhere[18]. Patients were recruited to the Post COVID-19 clinic, at the Karolinska University Hospital, a tertiary health care setting in Stockholm, Sweden, between July 2020 and December 2022. Patients hospitalised due to COVID-19 were routinely referred according to certain criteria (see inclusion criteria below) at discharge from different hospital departments, regardless of health status levels, as opposed to non-hospitalised individuals who were referred from primary care or self-referred due to persistent symptoms affecting working ability by 50% or more.

The following criteria were applied for inclusion in the study: 1) hospitalised patients, age ≥18, treated at ICU and/or at a ward requiring more than 2 l/min of oxygen, with extensive pulmonary radiographic changes and/or a complicated clinical course and/or 2) non-hospitalised patients referred from primary care with COVID-19-related symptoms from several organ systems with a previously verified or strongly suspected COVID-19 that had lasted for at least 3 months and impaired work ability by at least 50% or the corresponding impairment if not working. The exclusion criterion was digital visits only.

Ethical approval was granted by the Swedish Ethical Review Authority, and written informed consent was obtained from all participants (Dnr 2020-02149 with amendment 2020-04048, 2020-03215, 2020-06039, 2021-00988, 2022-02679-02, 2023-00982-02). The research was performed in accordance with the Declaration of Helsinki and relevant guidelines and regulations.

### Data sources and measurements

Data were collected at the participant´s first assessment at the Post COVID-19 clinic or retrieved from the patient´s medical records, consisting of demographics, medical history, date of acute infection and level of care, patient-reported persistent symptoms related to PCC and results from physical tests and self-reported questionnaires, in accordance with the previously published study protocol[18].

**Patient characteristics.** Demographic variables included age, sex, education, and occupational status. Health-related variables included body mass index (BMI), tobacco use, and comorbidities. Data were collected on the number of days of hospital stay as well as COVID-19-related treatment during the study period.

**Clinical outcomes.** Overall functional status was assessed with the Post-COVID-19 Functional Status scale (PCFS)[19], which consists of five items ranging from 0 (absence of any functional limitation) to 4 (severe functional limitations requiring assistance with activities of daily living). Participants were instructed to score their functional limitations prior to disease onset and at the actual visit, the latter covering the last week. Self-rated health was measured with the EQ visual analogue scale (EQ VAS), ranging from 0 to 100[20]. Dyspnoea was assessed with mMRC, consisting of five items ranging from 0 to 4, where a score ≥2 indicated a clinically significant dyspnoea. Additional variables included follow-up time, comprising days from illness onset until the first assessment at the actual visit at the Post COVID-19 clinic, as well as COVID-19 related variables, including sick leave, prevalent patient-reported symptoms related to PCC at follow-up and prevalence of postural orthostatic tachycardia syndrome (POTS), the most prevalent autonomic dysfunction phenotype[21].

**Physical function.** Self-reported physical activity was measured with the Frändin-Grimby activity scale, ranging from 1 to 6, with participants estimating their physical activity level in the last 2 weeks before disease onset and during the last week prior to the actual visit. Physical functioning was objectively assessed regarding exercise capacity using the Six-Minute Walk Test (6MWT). Lower extremity strength was measured with the 1-Minute Sit-To-Stand Test (1MSTST), where repetitions of completed stands for 60 seconds were recorded. Handgrip strength was measured with the Jamar hand-hold dynamometer, and results were recorded in kilograms (kg).

**Pulmonary function.** Lung function was objectively measured at the first clinical assessment with a spirometer (Medikro® Pro), including forced vital capacity (FVC), forced expiratory volume in one second ($FEV_1$), and the $FEV_1$/FVC ratio, following ATS/ERS guidelines[22]. Inspiratory muscle strength was measured with a hand-held device (MicroRPM) that measures maximal inspiratory pressure (MIP) in cm $H_2O$. All scores were compared to reference values and presented as a percentage of predicted values.

**Mental health, cognitive function and fatigue.** Depression was measured with the Patient Health Questionnaire-9 (PHQ-9), consisting of 9 items ranging from 0 to 3 to where participants scored their symptoms during the last two weeks, where a score ≥10 was considered indicative of clinical depression in the present study[23]. Anxiety was measured with the General Anxiety Disorder (GAD-7), consisting of 7 items ranging from 0 to 3, scoring the level of anxiety during the last two weeks, where a score ≥10 indicated anxiety[24]. Cognitive function was screened using the Montreal Cognitive Assessment (MoCA) test, where seven different cognitive domains are tested. A score <26 indicates cognitive impairment[25]. Physical fatigue was measured with the Fatigue Severity Scale (FSS), a questionnaire consisting of nine items. The participants

**Fig. 1 | Participant flow diagram of the study.**
Participant flow diagram of study population, including Non-Hospitalised and Hospitalised individuals referred to the Post COVID-19 clinic and then included in the ReCOV-project.

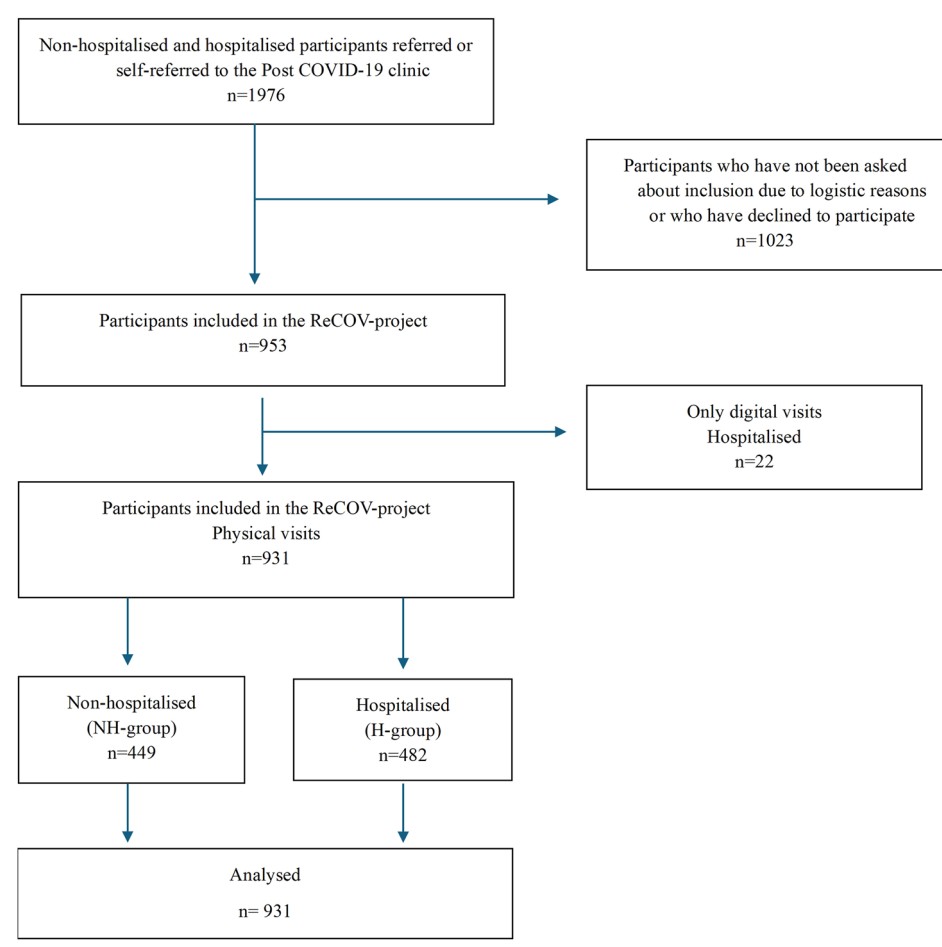

were instructed to score their level of physical fatigue during the last week on a 7-grade Likert scale, where a mean score ≥ 4 indicates fatigue[26].

## Statistical analyses

Data are presented as means and standard deviations for normally distributed continuous variables, as medians and interquartile ranges (IQR) for non-normally distributed numeric variables, and as frequencies ($n$) and percentages (%) for categorical data. The total sample consisted of $n = 931$ unique participants. Subsamples were used for analyses where data were only available for a subset (e.g., spirometry or repeated clinical visits); these sample sizes are reported in the respective Supplementary Data files and Supplementary Tables. Replicates in this study are defined as the individual study participants, each contributing a single observation per analysis.

The differences in clinical outcomes between the NH-group and the H-group are described with mean or median differences (MD) using a 95% confidence interval (CI). $P$-values were calculated as follows: for continuous variables, a two-sided $t$ test for normally distributed data, and otherwise, the Wilcoxon rank-sum test was applied. Chi-squared or Fisher's exact tests were used for categorical variables. Statistical significance was set at $\alpha = 0.05$. Statistical analyses were conducted using R version 4.3 (The R Foundation for Statistical Computing, Vienna, Austria).

**Regression analyses.** To examine the factors associated with self-rated health, EQ VAS was used as the dependent variable in a multivariable linear regression model. All clinically relevant variables (age, level of care, sex, days follow-up, mMRC, 6MWT % of predicted, MIP % of predicted, PHQ-9, PCFS, Frändin/Grimby, BMI, comorbidity, number of symptoms, sick leave, POTS diagnosis, MoCA, $SpO_2$ at rest, HR at rest, 1MSTST, % of predicted, grip strength % of predicted, FVC % of predicted, GAD-7, FSS) were included as independent variables without variable selection. To avoid multicollinearity, one of two highly correlated lung function measures was excluded; FVC (% predicted) was retained while $FEV_1$ was removed. Ordinal and continuous variables were categorised based on predefined clinical cut-offs.

For sensitivity analyses, we also tested dichotomised versions of PHQ-9 (≥ 10 vs. <10), GAD-7 (≥ 10 vs. <10), FSS (≥ 4 vs. <4), and MoCA (< 26 vs >26). All analyses were adjusted for age, level of care (non-hospitalised vs hospitalised), sex, and days of follow-up.

Missing data were addressed using multiple imputation by chained equations (MICE) with random forest models, using the mice package (v3.16.0) in R. Reproducibility of the regression results was strengthened by using multiply imputed datasets ($m = 30$), with results pooled according to Rubin's Rules[27], which provide stable estimates across repeated draws of missing data. The selected model had an adjusted R squared of 0.57.

**Clustering method.** A clustering analysis was conducted to identify distinct recovery profiles based on physical and mental health. Five clinically relevant variables were selected: PHQ-9, GAD-7, 6MWT, MIP, and the 1MSTST. To reduce dimensionality and account for correlations among variables, a principal component analysis (PCA) was performed on standardised values. Subsequently, a k-means clustering algorithm was applied using the first three principal components. A four-cluster solution was selected based on clinical interpretability, although the silhouette method suggested that two clusters would maximise separation.

## Results

A total of 1976 patients were referred or self-referred to the Post COVID-19 clinic. Of these, 1023 patients were excluded due to logistical reasons or because they declined participation. Consequently, 953 individuals consented to participate, of which 22 were excluded due to lack of follow-up data, i.e., no clinical assessment was available (Fig. 1). The final study sample consisted of 931 individuals (Fig. 1). Among the 931 participants, 449 (48%)

**Table 1 | A multivariable linear regression model investigating factors associated with self-perceived health EQ VAS presented as adjusted estimates and 95% confidence interval (CI), *n* = 931**

| | Estimate | 95% CI | *p* value |
|---|---|---|---|
| **Intercept** | 58.23 | −35.51 to 152.98 | 0.23 |
| mMRC ≥2 | −6.83 | −9.63 to −4.03 | 2.34e-06 |
| 6MWT ratio | 11.66 | 4.53 to 18.79 | 0.001 |
| Symptoms (number) | −0.65 | −0.95 to −0.36 | 1.78e-05 |
| Sick leave | −0.08 | −0.11 to −0.04 | 1.63e-05 |
| PHQ-9 depression ≥10 | −7.11 | −10.11 to −4.12 | 4.75-e06 |
| FSS fatigue (≥ 4) | −5.48 | −9.04 to −1.91 | 0.003 |
| MoCA cognitive impairment (<26) | −3.30 | −5.94 to −0.66 | 0.02 |
| Age | −0.03 | −0.15 to 0.08 | 0.59 |
| Level of care (hospitalised) | 8.34 | 4.07 to 12.61 | 0.0001 |
| Sex (male) | 2.15 | −1.50 to 5.80 | 0.25 |
| Follow-up | −0.003 | −0.01 to 0.004 | 0.41 |

All analyses were adjusted for age, level of care (not hospitalised vs hospitalised), sex, and days of follow-up. Ordinal and continuous variables were categorised based on predefined clinical cut-offs: mMRC ≥2 indicates a clinically significant dyspnoea. PHQ-9 ≥ 10 indicates a clinical depression; FSS ≥ 4 indicates fatigue. MoCA <26 indicates cognitive impairment. *mMRC* the modified Medical Research Council dyspnoea scale. *6MWT ratio* Six-Minute Walking Test ratio. *PHQ-9* the Patient Health Questionnaire-9, *FSS* Fatigue Severity Scale, *MoCA* Montreal Cognitive Assessment test.

were in the NH-group, 72% (670) contracted COVID-19 in the first wave, 19% (177) in the second wave, and 9% (84) in later waves. Most hospitalised individuals in our cohort were included during the first two waves of COVID-19 in Sweden, which were mainly caused by the original SARS-CoV-2 Wuhan strain (spring 2020) and the Alpha SARS-CoV-2 strain (fall and winter of 2020). The Delta SARS-CoV-2 strain did not evolve significantly in Sweden. Non-hospitalised individuals were included during 2020-2022, which means that they also were infected with the Omicron SARS-CoV-2 variants.

## Characteristics of the participants

The NH-group (age 44.4 ± 11.1 years) was comprised of 84% females, and comorbidities prior to COVID-19 (≥2) were present in 29.2% of the NH-group. In the H-group (age 56.7 ± 12.1 years), 28% were females and comorbidities prior to COVID-19 (≥2) were 52.3%. The average hospital stay for the H-group was 25.3 (±19.5) days, with 63.1% receiving intensive care and 43.8% requiring mechanical invasive ventilation. Characteristics of participants are presented in Table 1 (Supplementary Data 1).

## Clinical outcomes

The mean follow-up duration for the NH-group was 460 (±216) days post-initial COVID-19 illness, compared to 181 (±118) days for the H-group. An average of 59.6% in the NH-group and 27.4% in the H-group were on partial sick leave of 50% or more. On average, the NH-group reported a mean of 12 symptoms compared to a mean of 5 symptoms in the H-group. Symptoms for each group are displayed in Fig. 2 (source data in Supplementary Data 2). In the NH-group, 30.3% (*n* = 136) were diagnosed with POTS, compared to 2.5% (*n* = 12) in the H-group (Supplementary Data 3).

The NH-group demonstrated significantly lower values than the H-group in the following clinical outcome variables PCFS, EQ VAS, mMRC, Frändin-Grimby, 6 MWT, mental health and fatigue. While the H-group demonstrated significantly lower values than the NH-group in the following clinical outcomes variables: 1MSTS, lung function and cognitive function. Clinical outcomes at follow-up assessment are presented in Supplementary Data 3 and in Supplementary Table S1.

## Factors associated with self-rated health

In the final multivariable linear regression model, several clinical variables were significantly associated with self-rated health, measured by EQ VAS (Table 1).

Participants with a PHQ-9 score ≥10 reported on average 7.11 points lower EQ VAS compared to those with scores <10. (95% CI: −10.11 to −4.12 9.70). Similarly, having an FSS score <4 was associated with 5.48 points higher EQ VAS (95% CI: 1.85 to 9.11), and a 6MWT ratio was strongly positively associated with EQ VAS (estimate: +11,66.81 per unit; 95% CI: 4.53–18.79) (Table 1).

A clinically significant dyspnoea, as measured by mMRC ≥2, was associated with 6.83 points lower EQ VAS (95% CI: −963 to −4.04), and each additional symptom was associated with a −0.65 point decrease in EQ VAS (95% CI: −0.95 to −0.36). Having a normal MoCA score was associated with −3.30 point lower EQ VAS (95% CI: −5.94 to −0.66). Being on sick leave was also negatively associated with EQ VAS (−0.08 per percentage unit; 95% CI: −0.11 to −0.04). Hospitalisation was associated with 8.34 points higher EQ VAS scores (95% CI: 4.07 to 12.61) (Table 1).

## Cluster analysis

Based on the PCA, three dimensions were identified. The first two dimensions in the principal component analysis represented the psychological (PHQ-9 and GAD-7) and the physical (6MWT and 1MSTST) variables and the third dimension represented respiratory muscle strength (MIP). The k-means method was chosen because it yielded more clinically relevant clusters than the k-medoids method and latent profile analysis methods, which were also tested. While the silhouette method suggested that the optimal number of clusters was two, we opted for four clusters since we deemed them to be of greater clinical relevance, given the research questions. A total of 770 of the participants with complete data were included in the cluster analyses. Four main clusters, based on physical function and mental health, were identified: Cluster 1 (C1) included participants with very severe mental health issues and physical impairments, Cluster 2 (C2) was comprised of participants with severe mental health issues and moderate physical impairments, Cluster 3 (C3) included participants with moderate mental health issues and severe physical impairments, and Cluster 4 (C4) consisted of individuals with mild mental health issues and physical impairments (Fig. 3). Characteristics and clinical outcomes of the four clusters are displayed in Supplementary Data 4. Cluster 1 to 3 included 66.4% of all participants, varying from moderate to very severe mental health issues and physical impairments. The 10 most common symptoms in each cluster are described in Fig. 4. Detailed information on physical outcomes, pulmonary function, and mental health can be found in Supplementary Table S2. Prevalence of symptoms is presented in Supplementary data 5.

## Discussion

In this follow-up study, we present detailed data on characteristics and objectively measured persistent health impairments in a cohort comprised of either hospitalised or non-hospitalised patients with prior COVID-19. Approximately two-thirds of the study population exhibited significant physical and mental impairments during follow-up, with substantial inter-group differences. These impairments, together with self-reported symptoms, were also associated with a lower self-rated health. Upon further analysis, four distinct cluster profiles were identified based on mental and physical function.

Non-hospitalised and hospitalised individuals differed in demographic characteristics in agreement with previous findings[5]. The NH-group consisted of younger individuals of working age, predominantly women, who had been employed and physically active prior to initial COVID-19 illness and with few comorbidities. A majority in the NH-group were on sick leave at their first assessment, which had important implications for their own well-being and life economy[7]. In a recently published study [28], a hypothesis is presented suggesting that women of reproductive age may prioritise limited physiological resources toward reproductive functions, such as the

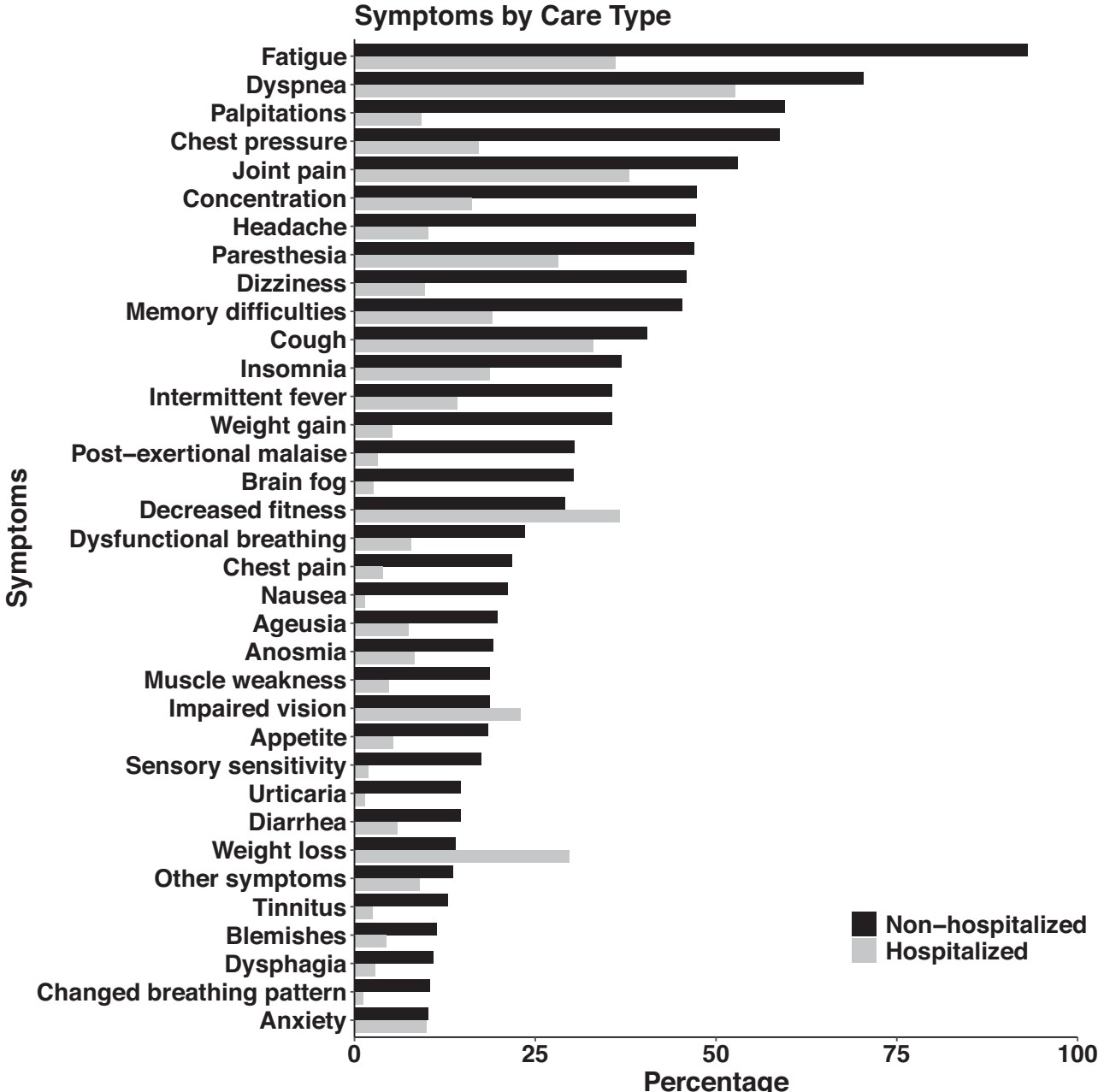

**Fig. 2 | All self-reported symptoms were presented by type of care.** All self-reported symptoms presented by type of care, Non-hospitalised (NH-group) (*n* = 449) and Hospitalised (H-group) (*n* = 482), with frequencies expressed as percentages.

menstrual cycle, potentially leading to suppression of parts of their immune system during SARS-CoV 2 infection. The first line of defence with type I interferons (IFN-I) is stronger in women and is normally sufficient to limit the severity of acute COVID-19. On the other hand, this strategy may pose an increased risk of viral persistence, leading to a higher risk of PCC in women. This hypothesis could also explain the more severe acute COVID-19 and cytokine storm in men, while women more often experience milder acute illness, but with an increased risk of PCC.

In line with other studies, the H-group was predominantly males with a mean age of 57 years, a higher BMI and a higher proportion of pre-existing comorbidities such as high blood pressure[29,30]. The H-group had relatively severe initial COVID-19 caused by the wild type, Alpha and Delta SARS-CoV-2 strains, with over 60% admitted to the ICU and of these 43% required invasive ventilation, which corresponds to similar proportions in other studies[1,30].

Overall, the participants presented a plethora of symptoms where the NH-group had a more complex symptom burden, whereas fatigue, dyspnoea and joint pain were common for both groups, in alignment with other studies[3,5]. However, the NH-group was referred from primary care or via self-referral due to symptoms with substantial impact on work ability, whereas the H-group was routinely invited for follow-up according to pre-specified criteria. This difference in referral pathways may have influenced the observed differences in symptom burden between the groups, with a higher symptom burden in the NH-group. A high proportion of individuals in both groups reported dyspnoea; however, the NH-group seemed to have more severe self-rated respiratory symptoms. The NH-group also demonstrated a more pronounced impairment in inspiratory muscle strength, while the H-group presented lower than predicted values in dynamic lung function measurements, indicating restrictive impairment. Studies of hospitalised patients have shown similar results regarding lung function[1,15].

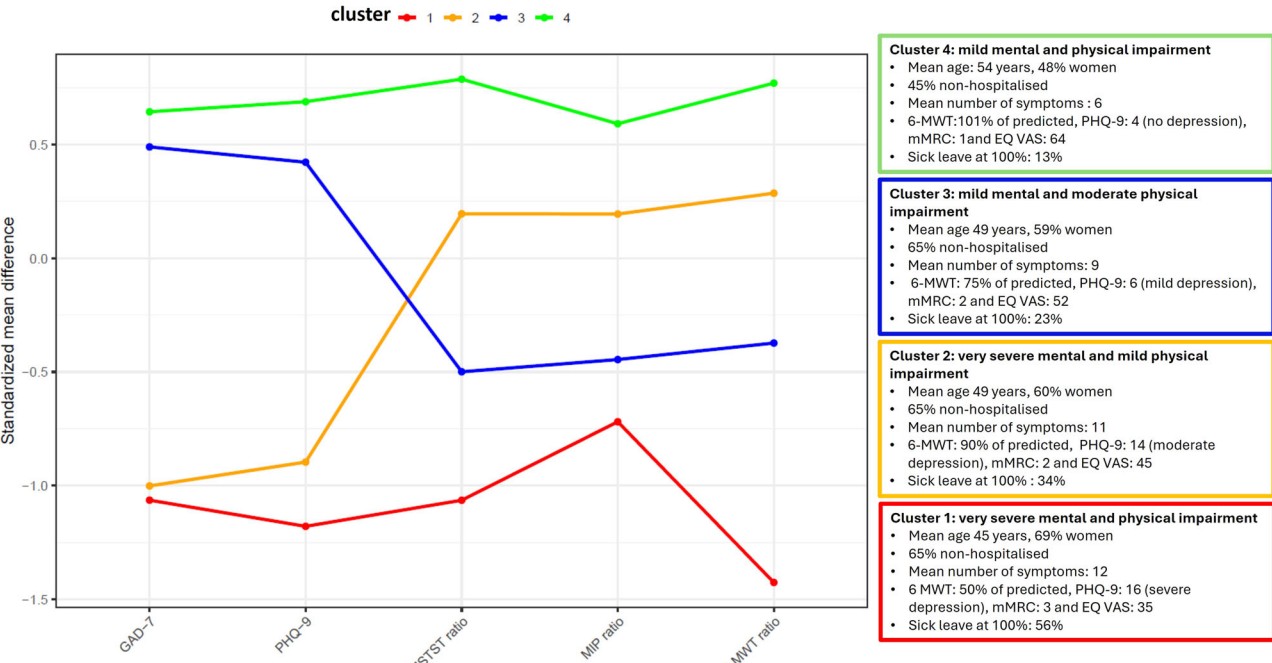

**Cluster 4: mild mental and physical impairment**
- Mean age : 54 years, 48% women
- 45% non-hospitalised
- Mean number of symptoms : 6
- 6-MWT:101% of predicted, PHQ-9: 4 (no depression), mMRC: 1and EQ VAS: 64
- Sick leave at 100%: 13%

**Cluster 3: mild mental and moderate physical impairment**
- Mean age 49 years, 59% women
- 65% non-hospitalised
- Mean number of symptoms: 9
- 6-MWT: 75% of predicted, PHQ-9: 6 (mild depression), mMRC: 2 and EQ VAS: 52
- Sick leave at 100%: 23%

**Cluster 2: very severe mental and mild physical impairment**
- Mean age 49 years, 60% women
- 65% non-hospitalised
- Mean number of symptoms: 11
- 6-MWT: 90% of predicted, PHQ-9: 14 (moderate depression), mMRC: 2 and EQ VAS: 45
- Sick leave at 100% : 34%

**Cluster 1: very severe mental and physical impairment**
- Mean age 45 years, 69% women
- 65% non-hospitalised
- Mean number of symptoms: 12
- 6 MWT: 50% of predicted, PHQ-9: 16 (severe depression), mMRC: 3 and EQ VAS: 35
- Sick leave at 100%: 56%

**Fig. 3 | Patterns of the different cluster profiles.** Visualising the pattern in segmentation within and between the different clusters profiles regarding mental (GAD-7, PHQ-9) and physical impairments (1 MSTS ratio, MIP ratio 6 MWT ratio) (n = 770). *GAD-7* General Anxiety Disorder −7, *PHQ-9* Patient Health Questionnaire, *1MSTS* One Minute Sit-To-Stand Test, *MIP* maximal inspiratory pressure, *6MWT* Six Minute Walking Test, *mMRC* (modified Medical Research Council) dyspnoea scale, *EQ VAS* EuroQol Visual Analogue Scale.

Residual radiologic findings and reduced diffusion capacity have been shown in previously hospitalised patients, which may be due to structural lung and vessel damage following pulmonary involvement in their initial case of COVID-19[13,31]. Non-hospitalised patients, on the other hand, demonstrated slightly different respiratory symptoms such as shortness of breath, chest pressure and a dysfunctional breathing pattern, which might be explained by non-structural pulmonary abnormalities such as air trapping, decreased lung perfusion and diaphragm dysfunction[3,8,32], with possible underlying cardiorespiratory dysautonomia, a still poorly defined condition[33]. Furthermore, many in the NH-group reported palpitations and notably around one third in this group were diagnosed with POTS compared to less than 3% in the H-group, as reported by others[33]. This concurs with female predominance in the NH-group, a typical feature of both PCC and POTS[17,21]. Fatigue was one of the most common symptoms and was strikingly more prevalent in the NH-group. Fatigue is a non-specific symptom, observed also in other post-infectious conditions, and its aetiology stretches beyond cardiorespiratory autonomic dysfunction, with some authors proposing viral persistence as the underlying cause[34]. Moreover, fatigue may be triggered by physical inactivity due to cardiorespiratory symptoms, and some individuals may also exhibit post-exertional malaise (PEM), which may lead to a vicious circle with further deconditioning[7].

In the present study, physical function was reduced regardless of hospitalisation, as shown by others[32]. Both the H- and NH-group reported joint pain as a common symptom, and the H-group also reported reduced fitness, which could depend on the initial severity of COVID-19 as well as a long hospital stay and PICS[15,16]. On the other hand, the NH-group had higher self-rated physical activity prior to COVID-19 compared to the H-group, with a lower self-rated value during follow-up, which indicates that the relative impact of post-COVID-19 sequelae may have been larger in the NH-group.

Depression (PHQ-9 ≥ 10) was more prevalent in the NH-group, amounting to 53.7% compared to the highest prevalence of depression after COVID-19 reported in other studies, which amounts to approximately 15%[35]. A study by Carlsson et al.[36] showed a general prevalence of 6% of depression in the general population in Stockholm County. The NH-group thus reported a substantially higher prevalence of depression than the general population; however, the latter group also reported a higher prevalence of mental health issues before initial COVID-19. Other studies show that self-reported anxiety and depression before COVID-19 are associated with a higher risk of developing PCC and are also associated with lower HRQoL[4,29]. Also, given that our sample included a higher proportion of women in the non-hospitalised group, it can be noted that menopause may further increase vulnerability to depression and anxiety[37]. A previous Swedish study has demonstrated that PCC is associated with worse mental health outcomes, which persist after 1 year[38].

Self-rated health (EQ VAS) was found to be low in the study cohort compared to normal values in Swedish population (EQ VAS: 76.1)[20] and was notably more pronounced in the NH-group. The low self-related health scores align with findings from other studies on PCC[4]. In our study, the regression analysis revealed independent associations between low self-rated health and symptoms such as fatigue, reduced physical capacity, depression, and dyspnoea. Patients on sick leave and patients with a high symptom burden also reported lower self-rated health. An interesting finding, albeit not entirely surprising, is that patients who have been hospitalised report better perceived health than non-hospitalised patients. These findings indicate that PCC poses a significant burden for the individual[7], which is comparable to other chronic conditions[39]. Non-hospitalised individuals comprise the majority of patients with PCC, which is a consequence that poses a significant burden also at a societal level[7].

We identified four distinct cluster profiles characterised by variations in physical functioning and mental health status. These profiles are essential for informing the development of targeted therapeutic and rehabilitative strategies tailored to individual recovery needs. Cluster 1 was characterised by a high symptom burden, including respiratory-related symptoms and palpitations. This cluster included both patients who required hospitalisation during the initial COVID-19 and those who did not, which underscores that also non-hospitalised individuals may have severe lingering symptoms and conditions with a significant impact on daily function[6]. Although initial COVID-19 presentation is not always associated with post-COVID-19

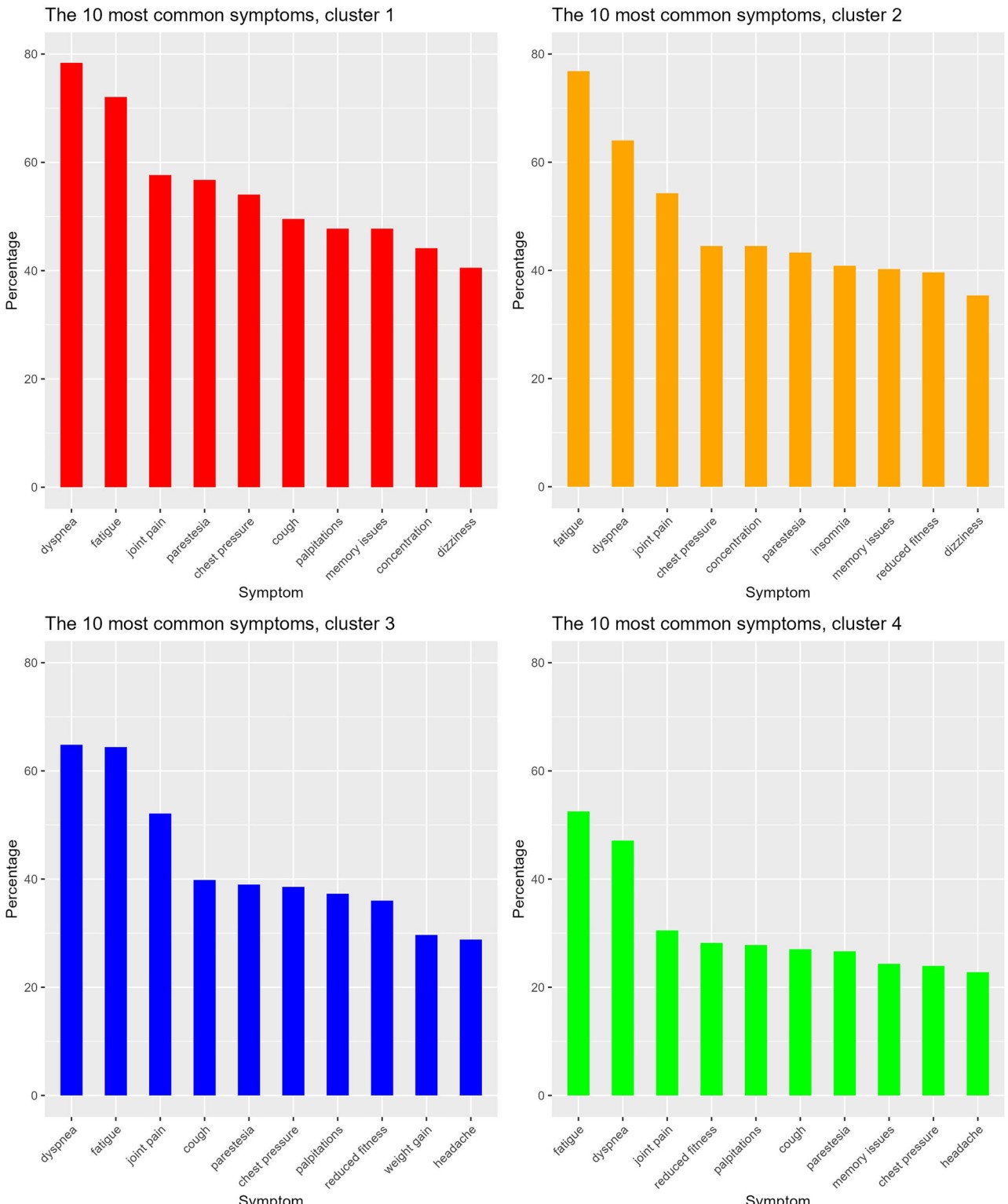

**Fig. 4 | The 10 most commonly self-reported symptoms.** The 10 most commonly self-reported symptoms presented by each cluster (Cluster 1 (*n* = 111), Cluster 2 (*n* = 164), Cluster 3 (*n* = 236), and Cluster 4 (*n* = 259)), with frequencies expressed as percentages.

symptom severity in non-hospitalised patients, a recent Scandinavian study reported that non-hospitalised individuals who were bedridden for more than 7 days at home were more likely to develop severe physical symptoms[40]. The prevalence of POTS, the most common post-COVID-19 dysautonomia, was prominent within this cluster, and the syndrome by itself often leads to severe impairments and a complex symptom burden[21]. Clusters 2 and 3, with either pronounced mental health impact or physical function

impairments, also exhibited relatively complex symptom profiles. In contrast, Cluster 4, which featured normal expected values for both physical function and mental health, reported higher self-rated health (EQ VAS: 63.6), but still lower compared to the general Swedish population[20]. This cluster constituted 34% of the cohort, and with a higher proportion of hospitalised participants (57%), indicating that they seemed to have recovered relatively well. A more effective clearance of viral particles could

be an explanatory factor for better outcomes in the H-group, as persistent viral remnants may trigger the immune system and have been proposed as an explanation for long-term symptoms and sequelae, particularly in non-hospitalised individuals[3,13,34].

Based on our results, we consider it crucial to use objective (e.g., physiological or performance-based tests) and subjective (e.g., self-report questionnaires) measurement tools to comprehensively assess physical and mental impairments in PCC to identify and prioritise different cluster profiles. Tailored interventions for individuals with more severe PCC can be achieved through multidisciplinary and multiprofessional teams to identify underlying conditions, the need for specific medical treatment and individually adjusted rehabilitation. Some of the measurements of physical function and mental health can be used as entry points for further diagnostic work-up, for example, in the identification of suspected PCC-POTS[33,41]. Further studies of long-term follow-up of recovery and the effect of targeted interventions for individuals with a high symptom burden, PCC, and more severe physical and mental impairments are urgently needed.

The strength of this study is the extensive and well-defined cohort with inclusion of both hospitalised and non-hospitalised individuals. However, the selection of participants towards the inclusion of more severe PCC, as they were referred to a specialist Post COVID-19 clinic, may not accurately reflect the broader population with milder PCC. We present detailed data derived from objective physical measurements, alongside self-reported information collected from questionnaires and medical records. We used normative reference values relevant for this group for the classification of physical impairment. The physical function outcome measures and the utilised questionnaires used in our study possess generic qualities that make them particularly appropriate for a highly heterogeneous patient population. As such, they have been highly recommended by multidisciplinary guidelines for managing patients with PCC, enhancing the generalisability of our findings[10,41,42].

Conversely, our study is not without limitations. The varying time frames of data collection, which aligned with different pandemic waves, different SARS-CoV-2 variants, and alterations to clinical routines, including assessments and tests, may have introduced variability within the cohort. We do, however, not judge the various SARS-CoV 2 variants to have an impact on outcome, as hospitalised individuals were included before the Omicron variants were present. In non-hospitalised individuals, some were infected also during the Omicron era, however other studies[43], including participants in Scandinavia have not shown differences in PCC severity. Approximately half of the eligible participants agreed to participate, which is due to logistical reasons and the fact that clinical routines changed over time. The first assessment of the NH-group did not occur until about a year and a half after their initial COVID-19 presentation, which was due both to a lack of knowledge regarding their need for a specialised assessment and a lack of available resources for this patient group, which at first was not at all known. Thus, we adjusted for the follow-up time within our multivariable models, aiming to control for potential discrepancies of the associations observed. These circumstances also contributed to missing data and a reduction in participant numbers for certain variables. The retrospective reporting of pre-COVID-19 functioning introduces potential for recall bias, as individuals may unintentionally over- or underestimate their prior health status. This limitation is particularly relevant in the context of a prolonged recovery period, where current symptom burden may influence perceptions of past functioning. While this bias is difficult to eliminate in retrospective designs, we believe that the use of robust and objective outcome measures, standardised self-report measures and the inclusion of a large and diverse sample help mitigate its impact to some extent. We also note that recall bias is a common challenge in post-illness recovery research, and we have taken care to interpret findings related to pre-COVID-19 functioning with appropriate caution.

Our study did not include a healthy control group or a cohort with less severe symptoms; however, we conducted comparisons against established reference values and clinically significant thresholds. Our inclusion criteria encompassed individuals with and without laboratory confirmation of COVID-19, a decision justified by the limited testing capacity in Sweden during the initial pandemic waves, which was largely restricted to hospitalised cases, with 72% in our cohort belonging to wave 1. However, the diagnosis of PCC was based on a well-substantiated clinical and epidemiological suspicion of COVID-19 and the trajectory of the illness according to the WHO definition[10].

## Conclusion

Our study shows that approximately two-thirds of patients with PCC, regardless of the level of care during their initial COVID-19 illness, exhibit varying degrees of persistent mental and physical impairment during follow-up after their initial illness, which can be shown in the different cluster profiles. There were substantial differences in demographic characteristics and symptom burden between hospitalised and non-hospitalised individuals, with older men with comorbidities in the former, and more women of reproductive age in the latter. Both groups were represented among those with more severe residual impairments and complex symptom panorama, but non-hospitalised patients had a higher proportion of cardiovascular dysautonomia, particularly POTS. These impairments and self-reported symptoms also negatively affected the self-rated health. Consequently, there is an urgent need for long-term studies investigating recovery trajectories and the development of effective, targeted interventions for individuals with post-COVID-19 condition.

## Author contributors

M.N.B., A.S.R., L.H., M.R.E., A.M.A., U.E., M.K., L.A., O.D., E.Å., K.C., M.S., M.R.U., J.B., E.R. contributed to the conceptualisation and study design. MNB and JB, as main applicants, were granted funding. A.S.R., A.S., A.T., C.H., J.Å.R., F.B. retrieved the data from medical records to an electronic database created by D.L., A.S.R. and M.B. Primarily, A.T. and A.S. reviewed and validated the data with assistance from A.S.R. and M.N.B. P.V. performed statistical analysis in collaboration with M.N.B., D.L., A.S.R., J.B. and E.R. All authors helped to interpret the findings. M.N.B., A.S.R., J.B. and E.R. wrote the initial draft and edited the following versions until submission of the manuscript. All authors revised the manuscript for critical content and approved the final version of the manuscript.

## Data availability

The source data for Fig. 2 are in Supplementary Data 2. The source data for Fig. 3 are in Supplementary Data 4. The source data for Fig. 4 are in Supplementary Data 5. Data is not publicly available, due to Swedish legislation, but is available upon request. However, metadata will be available through the KI Data Repository. Requests for access to the data can be put to our Research Data Office (rdo@ki.se) at Karolinska Institutet and will be handled according to the relevant legislation. This will require a data processing agreement or similar arrangement with the recipient of the data, as stated below. Conditions of access: Researchers must submit a written request outlining the intended use of the data, a brief research proposal, and documentation of ethical approval from a recognised ethics review board. Timeframe for response: all data access requests will be reviewed and responded to within 10 working days of receipt. Restrictions on use: The data may only be used for non-commercial academic research purposes. Redistribution of the data is not permitted. Access is subject to signing a data use agreement that outlines these conditions.

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

## Acknowledgements

The authors would like to thank all participants, data collectors, and colleagues at Karolinska University Hospital and Karolinska Institutet who contributed to performing the study. This study was supported by grants from: FORTE/FORMAS 2020–02789, The Swedish Heart Lung Foundation (2021-0024, 2021-0539, 2022-0563, 2021-0029, 2022-0224, 2020-0637,

2021-0536); The Swedish Research Council (2021-02844, 2022-00837). The study was also supported by Carl Bennet AB, Odd Fellows (B10) and Stiftelsen Bygg-Göta Vetenskaplig forskning och Social hjälpverksamhet. The funding sources have not been involved in any part of this manuscript.

## Funding

## Competing interests
M.N.B. reports receiving funding from the Swedish Research Council, the Swedish Heart-Lung Foundation, and FORTE/FORMAS. J.B. reports receiving funding from the Swedish Research Council, the Swedish Heart Lung Foundation, Stockholm Regional County and consulting fees from Novartis for lectures in the area as an expert in the field of PCC, as well as consulting fees from the Swedish Agency for Health Technology Assessment and Assessment of Social Services (SBU) for serving as an expert in the field. K.C. reports receiving support from the Swedish Heart Lung Foundation and Carl Bennet A.B. M.S. reports receiving funding from the Swedish Research Council, Pandemic Foundation, and Dysautonomia International. Consultant and consulting fees from the Swedish Agency for Health Technology Assessment and Assessment of Social Services (SBU) for serving as an expert in the field. The remaining authors declare no competing interests.

## Additional information

[1]Department of Neurobiology, Care Sciences and Society, Division of Physiotherapy, Karolinska Institutet, Stockholm, Sweden. [2]Women's Health, and Allied Health Professionals Theme, Medical Unit Allied Health Professionals, Karolinska University Hospital, Stockholm, Sweden. [3]Department of Clinical Neuroscience, Division of Psychology, Karolinska Institutet, Stockholm, Sweden. [4]Department of Women´s and Children´s Health, Karolinska Institutet, Stockholm, Sweden. [5]Department of Molecular Medicine and Surgery, Karolinska Institutet, Stockholm, Sweden. [6]Department of Clinical Physiology, Karolinska University Hospital, Stockholm, Sweden. [7]Department of Neurobiology, Care Sciences and Society, Division of Occupational Therapy, Karolinska Institutet, Stockholm, Sweden. [8]Department of Medicine Solna, Division of Infectious Diseases, Karolinska Institutet, Stockholm, Sweden. [9]Department of Clinical Science, Intervention and Technology, Division of Speech and language Pathology, Karolinska Institutet, Stockholm, Sweden. [10]Department of Neurobiology, Care Sciences and Society, Division of Clinical Geriatrics, Karolinska Institutet, Stockholm, Sweden. [11]Department of Clinical Science, Intervention and Technology, Karolinska Institutet, Stockholm, Sweden. [12]Department of Clinical Neuroscience, Centre for Psychiatry Research, Karolinska Institutet, & Stockholm Health Care Services, Region Stockholm, Stockholm, Sweden. [13]Jakobsberg University Primary Care Center, Stockholm Health Care Services, Region Stockholm, Stockholm, Sweden. [14]Department of Neurobiology, Care sciences and Society, Division of Nursing, Karolinska Institutet, Stockholm, Sweden. [15]Department of Perioperative Medicine and Intensive Care, Karolinska University Hospital, Solna, Stockholm, Sweden. [16]Division of Biostatistics Institute of Environmental Medicine, Karolinska Institutet, Stockholm, Sweden. [17]Department of Immunology & Inflammation, Imperial College London, London, UK. [18]Department of Medicine, Solna, Karolinska Institutet, Stockholm, Sweden. [19]Department of Cardiology, Karolinska University Hospital, Stockholm, Sweden. [20]Department of Respiratory Medicine and Allergy, Karolinska University Hospital, Stockholm, Sweden. [21]Department of Infectious Diseases, Karolinska University Hospital, Stockholm, Sweden. [22]These authors contributed equally: Judith Bruchfeld, Elisabeth Rydwik. ✉e-mail: malin.nygren-bonnier@ki.se

