## [Transparent Peer Review file · Communications Medicine]

Health outcomes in hospitalised and non-hospitalised individuals after COVID-19, an observational, cross-sectional study

Corresponding Author: Ms Malin Nygren-Bonnier

Version 0:

Reviewer comments:

Reviewer #1

(Remarks to the Author)

Dear Authors,

This article is highly relevant because it is a well-structured cohort study that monitors and evaluates the clinical, physical, and psychological consequences of COVID-19. Below, I present some questions that I had while reading and evaluating the article.

1 - A major issue in studies using self-reported tools, such as questionnaires, is the reliance on the participant's memory. It was not clear how clinical symptoms in non-hospitalized individuals were assessed. Did all individuals included in the study have a medical evaluation record from the period of COVID-19, or from a time close to the initial infection (whether from hospital, outpatient medical records, or primary care services)?

2 - An important variable regarding physical and psychological capacity is lung function. What motivated the choice of Maximum Inspiratory Pressure (MIP) for the clustering method? Additionally, why was spirometry not used in combination with MIP to construct the third dimension in the clustering model analysis?

3 - Another issue worth highlighting in relation to the assessment of mental health, cognitive function and self-assessed fatigue is the number of missing responses in the hospitalized group. Could this not cause a bias in relation to the results found?

4 - I missed a more in-depth discussion regarding the higher number of symptoms in the NH group (post-COVID-19) and the predominance of women (84%) in this group. Specifically, I would appreciate a discussion on whether the data might reflect a more pronounced physiological, hormonal, and functional decline in women over the age of 35 compared to men. Additionally, several studies suggest a higher percentage of depression diagnoses in women, and menopause itself may be a confounding variable, closely linked to diagnoses of anxiety and depression.

Reviewer #2

(Remarks to the Author)

Comments to the authors:

The manuscript entitled "Physical function, mental health and self-rated health in hospitalized and non-hospitalized individuals after COVID-19: A cross-sectional cohort study" primary aim was to describe symptoms and impairments of post-COVID-19 condition in hospitalized and non-hospitalized individuals and their association with self-rated health. I think this is worthwhile project, but I believe that there are some issues that need clarification. Listed below are my comments/questions to the authors of this manuscript.

Main critiques and questions for the authors:

1. A concern I have is that, while it seems the authors did valid scientific work and there are many interesting results presented, it remains unclear which findings are remarkable or novel. The discussion consists of too much repetition of the results, where interpretation of the findings is lacking. As a consequence, it is difficult to define what the major claims of this study are and what they add to our current knowledge. I would recommend shortening the discussion, focussing on clearly answering the research questions and placing these findings into context.
2. Another major concern is that I feel it is unsurprising that a large number of participants reported impairments during follow-up as participants were included from a post COVID-19 clinic, where the assumption is that people have persistent symptoms.
3. The terms objective and subjective measures are used to describe the methods used in this study. From the text it is difficult to assess which instruments are considered objective or subjective. Especially when assessing mental impairments, are validated questionnaires regarded as objective measures? If not, some claims in the discussion can not be made.
4. Some paragraphs in the methods section seem to be more results oriented than methods. For example, the description of the participant inclusion (line 152-156 and separately for each analysis), the sensitivity regression analyses performed (line 241-245), or result of the clustering are more fitting in the results. Keep in mind that anything that was not predefined (selections, models, clusters, components) should be presented in the results section, not the methods.
5. For me it is unclear what the difference is between the 951-22 included individuals, the analyzed cohort, the total cohort and the whole cohort (terms mentioned in the methods and results). Please use consistent language or be clear about the distinction. As different participant selections were used for the three research aims, this is especially important.
6. In the methods paragraph regression analyses it is mentioned there are variables with important missing data. This raises the question which data is missing and why, this is shortly mentioned in the discussion but could be covered earlier. Was imputation not an option to deal with missing data?
7. It is not the intention that the results section is a textual description of the variables in tables. Many outcomes described here could be removed if you refer to the tables. Now often differences between the NH and H groups are mentioned, but no test results are presented. Were statistical analyses performed that show whether these differences are significant?
8. Page 8 line 296: You mention that for all participants 5 symptoms are most prevalent. However, when looking at figure 2 there are big differences between the groups for which symptoms are most frequently reported. Reflection on this difference is missing. The confusion may be due to the clustering later in the manuscript, where differences between the clusters in reporting of symptoms are mentioned. If you present symptom reporting separately for the H and NH groups, I expect a reflection on the findings.
9. Retrospective reporting of pre-COVID functioning could lead to recall bias due to individuals' tendency to view past functioning with a distorted lens. This should be mentioned as a limitation of the study.
10. Discussion page 10, line 353: I am unsure if the statement that 'this is the first study to present detailed data ... patients' is founded. In recent years there have been many more studies published on these outcomes in COVID-19 cohorts, and even systematic reviews have been performed. While the authors have added "to our knowledge" I doubt this sentence should remain. What is new about this research?
11. Waiting periods to specialized PCC clinics are mentioned frequently in the discussion. However, it remains that there are few evidence-based treatments of PCC and it is still unclear which care these clinics can provide. So I feel it is too simplistic to say that specifically longer waiting periods lead to deconditioning, more mental health problems and lower self-rated health.
12. Page 13, line 495: In which way could selection bias of participants with milder PCC influence the results? Additionally, do the authors think there will also be a difference between self-referred and doctor/hospital referred participants?
13. There are quite a few mistakes regarding punctuation marks or grammar, please check all text for this. Also be consistent in the presentation of numbers, sometimes decimals are presented and other times not.

Minor critiques:

1. Summary: There is an I missing in the first mention of post-COVID-19 condition.
2. Page 8 line 298: I was searching for the symptom POTS in figure 2 but this is not presented there. Add that you can find the presence of POTS in table 2.
3. Figure 1: Split up the number of individuals excluded due to not being asked about inclusion and those who declined.
4. Figure 2: While percentages are now presented in the supplement, adding them to this figure could improve readability.
5. Captions are missing for figure 2 and 4.

Reviewer #3

(Remarks to the Author)

Thank you for the opportunity to review this interesting paper. Post-COVID-19 condition is still an important topic to research. This study reports findings that were well-executed and mostly well-presented. It also shows an interesting aspect, in that a previously non-hospitalised PCC group is seemingly worse off than a previously hospitalised group. However, I do have some major and minor critiques.

First of all, textually, the manuscript seems to still be in a draft stage. I recommend the authors to have the manuscript be checked by a native English speaker. Especially in the intro, some sentences are unnecessarily complicated and lengthy. There are also quite some spelling mistakes; some (non-exhaustive) examples: line 322: off instead of of, sometimes full stops are missing (line 406) or erroneously added (line 361)..., including even in the corresponding author information ('Physiotherpahy'). What are 'high resolution knowledge' (line 130) and 'self-related health' (line 427)? And why is one sentence repeated (lines 439 - 445)? Please perform spell checks (by text processor AND manual) before (re)submitting the manuscript.

Regarding the description of recruitment: were participants still undergoing treatment at the post-COVID clinic at the time of the baseline measurement? Could this have caused some biases in the recruitment (e.g. some form of self-selection) or interpretation? For example, it seems somewhat peculiar that patients were referred to this clinic while seemingly being mostly healthy (e.g. cluster 4 in your analysis).

Choices made for the regression model are untransparent. Can the authors explain why they have included these IV? Was collinearity assessed or expected? Why is the mMRC included but not the FSS, despite fatigue being more commonly and severely present in the sample? Same for the choice of the PHQ-9 over the GAD... Have you considered including a table with simple linear regression (or adjusted only for the control var) in the suppl. materials?

Finally, the purpose of the clustering analysis is unclear. Characteristics of the 4 clusters are presented, but without a clear aim. If this analysis aims to build towards risk profiles, why isn't a multinomial logistic regression performed, or even ANOVAs/Chi² tests? The discussion also doesn't really make clear why it is important to distinguish clusters 1, 2 and 3 from each other. While it's somewhat interesting to see that physical and mental health can differ in PCC patients (although not a novel finding), all groups still may - are even likely - to need multidisciplinary treatment approaches. The clustering analysis seems to be properly executed, but here too, transparency in choices can be improved. How have you decided on clinical relevance and how was this better for the K-means method over the other two? On a nitpicky note, no research questions (line 259) are presented in the manuscript, but research aims. It remains unclear to me what in the research aims makes it clear why 4 clusters would be preferable over (e.g.) 2.

Reviewer #4

(Remarks to the Author)

Major comments:

1. One major concern is that the statistical analyses only include participants with complete data and select variables based on the extent of missingness. This could result in significant information loss and bias, especially with approximately 30% of participants without complete data. It may be acceptable to exclude variables with extreme missingness (e.g., >80%). However, for variables with moderate missingness, it is critical to use appropriate imputation methods to handle missing data, which could reduce selection bias, preserve as much information as possible, and improve the robustness of the results. I recommend using multiple imputations in the analyses (see the two papers: Rubin, D. B. (1978). Multiple Imputations in Sample Surveys – A Phenomenological Bayesian Approach to Nonresponse; Rubin, D. B. (1978). Multiple Imputation for Nonresponse in Surveys).
2. The adjusted R squared of the final linear regression model for the EQ-VAS is 0.58, indicating that the model needs to be improved. Check for non-linearity and add spline terms (e.g., restricted cubic splines) for continuous predictors. Check the distribution of the EQ VAS and see if it is skewed. If so, consider using generalized linear models or semiparametric models (e.g., cumulative probability models).
3. The principal component analysis was used in the clustering analyses to reduce dimensions. The authors should provide a clear rationale for reducing the dimension and the proportion of variance explained by the chosen dimensions.
4. The manuscript seems to overlook different variants of the COVID-19 virus between 2020 and 2022. It may be helpful to look at the time (e.g., month) of the first COVID-19 diagnosis and adjust for it in the statistical analyses.
5. The time frame of recruiting participants in the abstract (June 2020 to December 2022) does not align with that in the main text line 149 Page 5 (July 2020 to December 2022). Please clarify this.
6. Provide time points of measurements for lung function in the Pulmonary function section.
7. Report p-values in Table 3

Minor comments:

1. Line 361 should be "upon further analysis, we identified four distinct clusters...."
2. Line 442, the sentence, "This cluster includes on daily function", is repeated.
3. Remove the red underscores in Figure 2.
4. The legend title of Figure 3 should be "Cluster" instead of "clust"
5. Align the inner lines in Table 4

Version 1:

Reviewer comments:

Reviewer #2

(Remarks to the Author)

I applaud the authors of this work for how much they have improved the manuscript based on the reviewer comments. They have responded mostly satisfactorily to my previous comments and questions. Important references and explanations have

been added, and (description of) statistical analyses have been improved. Listed below are my final comments to the authors of this manuscript.

1. In my opinion the description of differences between self-referred and doctor/hospital referred participants in the manuscript is still lacking. Especially when claims such as "The NH-group reported significantly more symptoms compared to the H-group" are made in the plain language summary, it is essential to understand that hospitalized patients were followed up regardless of symptom persistence whereas nonhospitalized individuals were more likely to seek medical attention only if symptoms persisted, which the authors nicely pointed out in the response to reviewer comments.

2. I feel conflicted about the naming of the cluster groups as "recovery profiles". Since this study was cross-sectional it did not study recovery of PCC, only the presence of persisting symptoms. No health status prior to COVID-19 or in the acute phase of disease was included with the exception of comorbidities. Referring to recovery profiles gives the perception that the study was longitudinal.

Reviewer #4

(Remarks to the Author)

Reviewer #5

(Remarks to the Author)

Response to R#3

The comments looks good to me, but I think the paragraphs about regression analyses need to be clearer:

1. Please state the purpose of doing regression analysis, for example "To examine the factors associated with self-rated health" or something else. It is now not very clear what the analysis is for.

2. Line 247 "All clinically relevant variables" is not clear, please list out the variables included in the model.

3. In the method section, it says that "Interaction terms between level of care (hospitalised vs. non-hospitalised) and selected covariates (dyspnoea, physical function, mental health, cognition, symptoms, and depression) were included in the main model" and "Based on the hypothesis that associations between clinical variables and EQ VAS may differ by follow-up time, level of care, or comorbidity burden, the model was extended to include relevant interaction terms.

". However, in the table of regression results (Table 3), no results for interaction terms was reported. Please either revise the methods or the result table to make them align with each other.

4. In the result section line 308 to 325, please signpost to the result table accordingly.

Version 2:

Reviewer comments:

Reviewer #2

(Remarks to the Author)

The authors have responded to all comments and have adjusted the manuscript accordingly. I compliment them on great work.

Reviewer #5

(Remarks to the Author)

The concerns have been addressed and I don't have further comments.

Dear Editor and Reviewers,

Thank you for your thorough and insightful feedback. Your input provides an important contribution to enriching and improving the manuscript. We appreciate the time and effort you have invested in reviewing our work. Your suggestions have significantly enhanced the clarity and robustness of our study, and we have made the necessary revisions accordingly. We are grateful for your valuable contributions, which have helped us to refine our research and present a more comprehensive and accurate analysis.

Best regards, corresponding author

Comment from the reviewers	Responses from the authors
Editor 1) include major claims and contributions, as well as research objectives, (2) include updated results after the data missingness has been addressed, (3) comment on the validity (potential bias towards patients from a specific post-COVID clinic, in particular), (4) include improved regression and clustering analyses, and report statistical significance of the results, 5) spellcheck 6) Additionally, we want to stress the importance of being transparent and making all limitations clear when revising the manuscript.	Thanks for all valuable comments, we have now made the suggested revisions: 1) This has now been revised and clarified 2) This has been addressed and updated. 3) This has now been added in strength and limitations 4) The regression analyses have been revised and improved, and the cluster analysis has been further clarified 5) The manuscript has been revised by an authorized language firm 6) This has been clarified
Reviewer 1 A major issue in studies using self-reported tools, such as questionnaires, is the reliance on the participant's memory. It was not clear how clinical symptoms in non-hospitalized individuals were assessed. Did all individuals included in the study have a medical evaluation record from the period of COVID-19, or from a time close to the initial infection (whether from hospital, outpatient medical records, or primary care services)?	Thank you for your comment. We have clarified this in the methods section. All patients were asked to report any persistent symptoms during their first clinical visit. These symptoms were recorded in their medical records, and the data was subsequently collected into our database. We have also discussed self-reported tools and reliance of the participants memory under limitations. Please see, page 5, line 175 in the marked-up manuscript: “Data was collected at the participant’s first assessment at the Post COVID-19 clinic or retrieved from the patient’s medical records consisting of demographics, medicals history, date of acute infection and level of care, patient reported persistent symptoms....” Please see, page 12, line 485-487 “. The retrospective reporting of pre-COVID-19 functioning introduces potential for recall bias, as individuals may unintentionally over- or underestimate their

	prior health status. This limitation is particularly relevant in the context of a prolonged recovery period, where current symptom burden may influence perceptions of past functioning. While this bias is difficult to eliminate in retrospective designs, we believe that the use of robust and objective outcome measures, standardised self-report measures and the inclusion of a large and diverse sample help mitigate its impact to some extent. We also note that recall bias is a common challenge in post-illness recovery research, and we have taken care to interpret findings related to pre-COVID-19 functioning with appropriate caution.”
An important variable regarding physical and psychological capacity is lung function. What motivated the choice of Maximum Inspiratory Pressure (MIP) for the clustering method? Additionally, why was spirometry not used in combination with MIP to construct the third dimension in the clustering model analysis?	Thank you for your important comment. We chose Maximum Inspiratory Pressure (MIP) instead of lung function measures because our preliminary results, as well as the clinical context, indicated that MIP was more relevant. Furthermore, MIP was used as an objective measurement of diaphragm muscle-strength, which, based on our clinical data, initially appeared to be more affected than lung function. Additionally, we aimed to base our clustering on the largest possible number of participants. Therefore, we selected a single measure of respiratory function.
Another issue worth highlighting in relation to the assessment of mental health, cognitive function and self-assessed fatigue is the number of missing responses in the hospitalized group. Could this not cause a bias in relation to the results found?	Thank you for your very relevant comment. As this could indeed cause bias, we have now performed an imputation and also improved the regression analyses (see sections of statistical analyses and results).
I missed a more in-depth discussion regarding the higher number of symptoms in the NH group (post-COVID-19) and the predominance of women (84%) in this group. Specifically, I would appreciate a discussion on whether the data might reflect a more pronounced physiological, hormonal, and functional decline in women over the age of 35 compared to men. Additionally, several studies suggest a higher percentage of depression diagnoses in women, and menopause itself may	Thank you for a very interesting question. One of our co-authors, Professor Brodin, recently published an article in Nature discussing this, so we have added: at page 9, line 355 – 363. In a recently published study, a hypothesis is presented suggesting that women of reproductive age may prioritise limited physiological resources toward reproductive functions, such as the menstrual cycle, potentially leading to suppress parts of their immune system during SARS-CoV 2 infection. The first line of defence with type I interferons (IFN-I) is stronger in women and is normally sufficient to limit the severity of acute COVID-19. On the other hand, this strategy may pose an increased risk of viral persistence, leading to a higher risk of PCC in women. This hypothesis could also explain the more severe acute COVID-19 and cytokine storm in men, while women more often experience milder acute illness but with an increased risk of PCC. Reference: Lakshmikanth T,Brodin P. Immune system adaptation during gender-affirming testosterone treatment. Nature. 2024 Sep;633(8028):155-164. doi: 10.1038/s41586-024-07789-z. Epub 2024 Sep 4. Erratum in: Nature. 2024 Oct;634(8033):E5. doi: 10.1038/s41586-024-08081-w. PMID: 39232147; PMCID: PMC11374716.

be a confounding variable, closely linked to diagnoses of anxiety and depression.	Thank you for your thoughtful comment concerning depression. We appreciate the opportunity to further discuss the predominance of women in the non-hospitalized group. As you noted, hormonal factors especially in women over 35, may play a role. Perimenopausal and menopausal transitions are associated with changes in oestrogen levels, which can influence mood. Although our dataset lacks hormonal measures, we acknowledge their potential impact and have expanded our discussion to emphasize this: At page 10, line 401-412): Depression was more prevalent in the NH-group, 53.7 % as compared to the highest prevalence of depression after COVID-19 reported in other studies which amounts to approximately 15 %. A study by Carlsson et al showed a general prevalence of 6 % of depression in the general population in Stockholm County. The NH-group thus reported a substantially higher prevalence of depression than the general population, however this group also reported a higher prevalence of mental health issues before initial COVID-19. Other studies show that self-reported anxiety and depression before COVID-19 are associated with a higher risk of developing PCC and are also associated with a lower HRQoL. Also, given that our sample included a higher proportion of women in the non-hospitalized group, it can be noted that menopause may further increase vulnerability to depression and anxiety. Reference: Alblooshi, S., Taylor, M., & Gill, N. (2023). Does menopause elevate the risk for developing depression and anxiety? Results from a systematic review. Australasian psychiatry: bulletin of Royal Australian and New Zealand College of Psychiatrists, 31(2), 165–173. https://doi.org/10.1177/10398562231165439
Reviewer 2 A concern I have is that, while it seems the authors did valid scientific work and there are many interesting results presented, it remains unclear which findings are remarkable or novel. The discussion consists of too much repetition of the results, where interpretation of the findings is lacking. As a consequence, it is difficult to define what the major claims of this study are and what they add to our current knowledge. I would recommend shortening the discussion, focusing on clearly answering the research questions and placing these findings into context.	Thank you for your valuable feedback. We have now conducted a thorough review of the discussion section, where we have tried to highlight the key messages, their significance, and how they address the research questions of the study. We have also removed repetitions and shortened certain parts. We hope that these changes in the revised are satisfactory.

Another major concern is that I feel it is unsurprising that a large number of participants reported impairments during follow-up as participant were included from a post COVID-19 clinic, where the assumption is that people have persistent symptoms.	Thanks for your comment, at the time of the study little was known regarding persistent symptoms and impairments both in hospitalized and non-hospitalized individuals. The study was exploratory, based on data from a thorough clinic work-up. The study adds novel information regarding objectively measured testing methods, not only symptoms. This is further clarified in the discussion, page 9, line.343-345. “In this follow-up study we present detailed data on characteristics and objectively measured persistent health impairments in a cohort comprising either hospitalised or non-hospitalised patients with post-COVID-19 sequalae.”
The terms objective and subjective measures are used to describe the methods used in this study. From the text it is difficult to assess which instruments are considered objective or subjective. Especially when assessing mental impairments, are validated questionnaires regarded as objectives measures? If not, some claims in the discussion can not be made.	Thank you for this insightful comment. We acknowledge that our definitions of objective and subjective measures were not sufficiently clear. In this study, we define 'objective measures' as assessments independent of self-report, such as biological markers or performance-based tests. 'Subjective measures' refer to self-reported data, including validated questionnaires assessing psychological symptoms. While validated questionnaires are standardized and widely used in clinical research, they still rely on participants' self-assessment and are therefore categorized as subjective in our study. We have carefully reviewed the discussion section to ensure that claims related to objective and subjective measures are appropriately framed. To improve clarity, we have revised the discussion to ensure consistency with these definitions: Page 11, line 448-451: Based on our results, we consider it crucial to use objective (e.g., physiological or performance-based tests) and subjective (e.g., self-report questionnaires) measurement tools to comprehensively assess physical and mental impairments in PCC in order to identify and prioritise different recovery profiles. Additionally, in the methods section (p. 6), we added references to the questionnaires under Mental health, cognitive function, and fatigue, for transparency.
Some paragraphs in the methods section seem to be more results oriented than methods. For example, the description of the participant inclusion (line 152-156 and separately for each analysis), the sensitivity regression analyses performed (line 241-245), or result of the clustering are more fitting in the results. Keep in mind that anything that was not predefined (selections, models, clusters, components) should be presented in the results section, not the methods.	Thank you for the comment and this is now moved to the result section as suggested.

For me it is unclear what the difference is between the 951-22 included individuals, the analyzed cohort, the total cohort and the whole cohort (terms mentioned in the methods and results). Please use consistent language or be clear about the distinction. As different participant selections were used for the three research aims, this is especially important.	Thank you for noticing inconsistencies in the terminology. We have thoroughly revised the manuscript and have streamlined terminology according to your comment.
In the methods paragraph regression analyses it is mentioned there are variables with important missing data. This raises the question which data is missing and why, this is shortly mentioned in the discussion but could be covered earlier. Was imputation not an option to deal with missing data?	Thank you for this important comment. In response, we revised the analysis to fully address the issue of missing data. We applied a multiple imputation strategy using the R package mice, with 30 imputed datasets, based on a random forest algorithm and 1000 iterations for convergence. Imputed datasets were analysed separately, and results were combined using Rubin's rules through pool, standard MICE procedure in R.
It is not the intention that the results section is a textual description of the variables in tables. Many outcomes described here could be removed if you refer to the tables. Now often differences between the NH and H groups are mentioned, but no test results are presented. Were statistical analyses performed that show whether these differences are significant?	Thank you for this comment, we have now adjusted the result section as suggested. P-values are now calculated as follows: for continuous variables, a t-test was used if the variable was judged to be approximately normally distributed based on visual inspection and summary statistics (e.g., EQ-5D, MIP, 6MWT, FVC, etc.); otherwise, the Wilcoxon rank-sum test was applied. For categorical variables, a Chi-squared test was used unless expected cell counts were low, in which case Fisher's exact test was employed. A p-value < 0.05 was considered statistically significant.
Page 8 line 296: You mention that for all participants 5 symptoms are most prevalent. However, when looking at figure 2 there are big differences between the groups for which symptoms are most frequently reported. Reflection on this difference is missing. The confusion may be due to the clustering later in the manuscript, where differences between the clusters in reporting of symptoms are mentioned. If you present symptom reporting separately for the H and NH groups, I expect a reflection on the findings.	Thank you for highlighting this point. The explorative report in this matter is now presented in the result section, and a brief reflection on the differences in symptom presentation between the groups has been added to the discussion section.
Retrospective reporting of pre-COVID functioning could lead to recall bias due to individuals' tendency to view past functioning with a distorted lens. This should be mentioned as a limitation of the study.	We agree that the retrospective reporting of pre-COVID functioning introduces potential for recall bias, as individuals may unintentionally over- or underestimate their prior health status. This limitation is particularly relevant in the context of a prolonged recovery period, where current symptom burden may influence perceptions of past functioning. In response to this, we have now explicitly acknowledged this limitation in the revised manuscript. While this bias is difficult to eliminate in retrospective designs, we believe that the use of standardized self-report measures and the inclusion of a large and diverse sample help mitigate its

	impact to some extent. We also note that recall bias is a common challenge in post-illness recovery research, and we have taken care to interpret findings related to pre-COVID functioning with appropriate caution.
Discussion page 10, line 353: I am unsure if the statement that ‘this is the first study to present detailed data ... patients’ is founded. In recent years there have been many more studies published on these outcomes in COVID-19 cohorts, and even systematic reviews have been performed. While the authors have added “to our knowledge” I doubt this sentence should remain. What is new about this research?	Thanks for this valuable comment, this has been rewritten. Page 9, line 343-345
Waiting periods to specialized PCC clinics are mentioned frequently in the discussion. However, it remains that there are few evidence-based treatments of PCC and it is still unclear which care these clinics can provide. So I feel it is too simplistic to say that specifically longer waiting periods lead to deconditioning, more mental health problems and lower self-rated health.	We acknowledge that evidence-based treatments for PCC remain limited, and the part regarding longer waiting periods is now removed.
Page 13, line 495: In which way could selection bias of participants with milder PCC influence the results? Additionally, do the authors think there will also be a difference between self-referred and doctor/hospital referred participants?	We thank the reviewer for raising this important point. As described in the Methods section, patients who had been hospitalized for COVID-19 were automatically referred to the post-COVID outreach clinic based on predefined inclusion criteria, primarily reflecting the severity of their illness. In contrast, non-hospitalised patients were referred by primary care physicians due to persistent symptoms. This difference in referral pathways introduces a potential source of bias. Hospitalised patients were followed up regardless of symptom persistence, often motivated by a desire to understand their recovery status or to receive reassurance, even in the absence of ongoing symptoms. In contrast, non-hospitalized individuals were more likely to seek medical attention only if symptoms persisted, potentially after a period of waiting for spontaneous recovery. This may have led to some degree of deconditioning in this group, which could influence both their symptom burden and functional status at follow-up. Page 11, line 460-463 “The strength of this study is the extensive and well-defined cohort with inclusion of both hospitalised and non-hospitalised individuals. However, the selection of participants towards the

	inclusion of more severe PCC as they were referred to a specialist post-COVID clinic may not accurately reflect the broader population with milder PCC.”
There are quite a few mistakes regarding punctuation marks or grammar, please check all text for this. Also be consistent in the presentation of numbers, sometimes decimals are presented and other times not.	The manuscript has undergone authorized language editing.
Minor critiques:  1. Summary: There is an I missing in the first mention of post-COVID-19 condition. 2. Page 8 line 298: I was searching for the symptom POTS in figure 2 but this is not presented there. Add that you can find the presence of POTS in table 2. 3. Figure 1: Split up the number of individuals excluded due to not being asked about inclusion and those who declined. 4. Figure 2: While percentages are now presented in the supplement, adding them to this figure could improve readability. 5. Captions are missing for figure 2 and 4. 	Thank you for these suggestions. We have now revised the text accordingly.  1. Changed as suggested 2. This is now added 3. As this study is based on data from an outpatient clinic, there was a high patient turnover, and the clinic's routines and staff also changed. Unfortunately, we are unable to break down this figure as we did not keep detailed records of the number of patients who declined participation or whom we missed asking. Therefore, the total number represents all those we know attended the clinic but was not included. 4. We initially attempted to include this information in Figure 2. However, it was difficult to read the figure with the percentages included, as they became too small. Therefore, we have placed this information in the supplementary table. 5. This has now been added and are placed at the end of the main manuscript
Reviewer 3	
First of all, textually, the manuscript seems to still be in a draft stage. I recommend the authors to have the manuscript be checked by a native English speaker. Especially in the intro, some sentences are unnecessarily complicated and lengthy. There are also quite some spelling mistakes; some (non-exhaustive) examples: line 322: off instead of of, sometimes full stops are missing (line 406) or erroneously added (line 361)...., including even in the corresponding author information ('Physioterpahy'). What are 'high resolution knowledge' (line 130) and 'self-related health' (line 427)? And why is one sentence repeated (lines 439 - 445)? Please perform spell checks (by text processor AND manual) before (re)submitting the manuscript.	The manuscript has undergone authorized language editing.
Regarding the description of recruitment: were participants still undergoing treatment at the post-COVID clinic at the time of the baseline measurement? Could this have caused	As described in the Methods section, patients who had been hospitalized for COVID-19 were automatically referred to the post-COVID outreach clinic based on predefined inclusion criteria. This

some biases in the recruitment (e.g. some form of self-selection) or interpretation? For example, it seems somewhat peculiar that patients were referred to this clinic while seemingly being mostly healthy (e.g. cluster 4 in your analysis).	referral was primarily driven by the severity of their illness, which necessitated inpatient care and thus warranted structured follow-up. In contrast, non-hospitalized patients were referred by primary care physicians based on the presence of persistent symptoms, as outlined in the inclusion criteria. It is important to note that several hospitalised patients demonstrated substantial recovery by the time of follow-up, which is reflected in their representation in Cluster 4—characterized by good physical and mental health. Additionally, some non-hospitalized individuals experienced a relatively long delay before being referred for follow-up. This delay may have allowed for spontaneous symptom resolution or improvement, potentially explaining why some of these patients also appeared in Cluster 4. These observations underscore the heterogeneity of recovery trajectories and support the value of cluster analysis in capturing clinically meaningful variation beyond initial disease severity.
Choices made for the regression model are untransparent. Can the authors explain why they have included these IV? Was collinearity assessed or expected? Why is the mMRC included but not the FSS, despite fatigue being more commonly and severely present in the sample? Same for the choice of the PHQ-9 over the GAD... Have you considered including a table with simple linear regression (or adjusted only for the control var) in the suppl. materials?	We appreciate the reviewer’s comment. In response, we revised the analysis to fully address the issue of missing data. We applied a multiple imputation strategy using the R package mice, with 30 imputed datasets, based on a random forest algorithm and 1000 iterations for convergence. Imputed datasets were analyzed separately, and results were combined using Rubin’s rules through pool, standard MICE procedure in R. Given the exploratory nature of the study and the application of multiple imputations to address missing data, we decided to include all available clinical outcomes in the analysis (for example FSS, GAD, etc). This approach was taken to ensure a comprehensive and unbiased evaluation of potential associations, in line with the study’s hypothesis-generating objectives. Regarding collinearity we also checked multicollinearity (VIFs), noting that while multicollinearity naturally increases with interaction terms, no concerning patterns were observed.

The purpose of the clustering analysis is unclear. Characteristics of the 4 clusters are presented, but without a clear aim. If this analysis aims to build towards risk profiles, why isn't a multinomial logistic regression performed, or even ANOVAs/Chi² tests? The discussion also doesn't really make clear why it is important to distinguish clusters 1, 2 and 3 from each other. While it's somewhat interesting to see that physical and mental health can differ in PCC patients (although not a novel finding), all groups still may - are even likely - to need multidisciplinary treatment approaches. The clustering analysis seems to be properly executed, but here too, transparency in choices can be improved. How have you decided on clinical relevance and how was this better for the K-means method over the other two? On a nitpicky note, no research questions (line 259) are presented in the manuscript, but research aims. It remains unclear to me what in the research aims makes it clear why 4 clusters would be preferable over (e.g.) 2.	A cluster analysis was conducted to identify clinically meaningful recovery profiles based on both physical and mental health indicators. Five variables (PHQ-9, GAD-7, 6MWT % predicted, MIP % predicted, and 1MSTST % predicted) were selected a priori for their clinical relevance in post-COVID-19 rehabilitation. Missing values were excluded (complete-case analysis). Principal component analysis (PCA) was used to reduce dimensionality and remove collinearity, with the first three principal components included in the clustering. A k-means algorithm was applied using four clusters (k = 4), chosen based on clinical interpretability and visual inspection, although the silhouette method suggested k = 2. Cluster identities were re-ordered based on functional status for interpretability (1 = poorest health, 4 = best). Mean standardized scores were used to illustrate the characteristics of each cluster. The presentation of four clusters was therefore considered more informative and clinically meaningful, particularly in the context of developing precision-based interventions tailored to specific recovery trajectories. By describing/exploring the characteristics and outcomes associated with each cluster, we aimed to generate clinically meaningful insights that could inform more tailored, precision-based care strategies. Therefore, the cluster analysis itself served as the analytical framework to differentiate these subgroups. In line with this objective, we did not perform additional inferential statistical tests such as chi-square, ANOVA, or multinomial regression, as our goal was descriptive rather than hypothesis-testing. We agree with the reviewer that multidisciplinary care is often essential in post-COVID recovery. However, identifying whether an individual presents with predominantly mental or physical impairments may help guide more targeted interventions. In this context, the cluster analysis provides a valuable tool for stratifying patients and supporting individualized rehabilitation planning.
Reviewer 4 One major concern is that the statistical analyses only include participants with complete data and select variables based on the extent of missingness. This could result in significant information loss and bias, especially with approximately 30% of participants without complete data. It may be acceptable to exclude variables with extreme missingness (e.g., >80%).	Thank you for this important comment. In response, we revised the analysis to fully address the issue of missing data. We applied a multiple imputation strategy using the R package mice, with 30 imputed datasets, based on a random forest algorithm and 1000 iterations for convergence. Imputed

However, for variables with moderate missingness, it is critical to use appropriate imputation methods to handle missing data, which could reduce selection bias, preserve as much information as possible, and improve the robustness of the results. I recommend using multiple imputations in the analyses (see the two papers: Rubin, D. B. (1978). Multiple Imputations in Sample Surveys – A Phenomenological Bayesian Approach to Nonresponse; Rubin, D. B. (1978). Multiple Imputation for Nonresponse in Surveys).	datasets were analyzed separately, and results were combined using Rubin’s rules through pool, standard MICE procedure in R.
The adjusted R squared of the final linear regression model for the EQ-VAS is 0.58, indicating that the model needs to be improved. Check for non-linearity and add spline terms (e.g., restricted cubic splines) for continuous predictors. Check the distribution of the EQ VAS and see if it is skewed. If so, consider using generalized linear models or semiparametric models (e.g., cumulative probability models).	Thank you for this thoughtful comment. We agree that the adjusted R² of approximately 0.57 indicates that the model explains a moderate portion of the variance. In this study, we opted for a standard linear regression model to prioritize interpretability. We acknowledge that we did not explore non-linear effects or alternative model structures (e.g., splines or generalized linear models). We also checked multicollinearity (VIFs), noting that while multicollinearity naturally increases with interaction terms, no concerning patterns were observed.
The principal component analysis was used in the clustering analyses to reduce dimensions. The authors should provide a clear rationale for reducing the dimension and the proportion of variance explained by the chosen dimensions.	PCA reduced data dimensionality and captured the main variation in physical and mental health variables. This allowed for clustering based on combined underlying profiles, rather than individual metrics, improving the robustness and interpretability of the subgroup classification.
The manuscript seems to overlook different variants of the COVID-19 virus between 2020 and 2022. It may be helpful to look at the time (e.g., month) of the first COVID-19 diagnosis and adjust for it in the statistical analyses.	Thank you for the relevant comment regarding different variants of the COVID-19 virus. The majority of hospitalised individuals in our cohort were included during the first two waves of Covid-19 in Sweden which were mainly caused by the original SARS Cov-2 Wuhan strain (spring 2020) and the alfa SARS Cov-2 strain (fall and winter of 2020). The delta SARS Cov-2 strain did not evolve significantly in Sweden. Non-hospitalised individuals were included during 2020-2022 which means that they also were infected with the Omcron SARS CoV-2 Variants. In the analyses we adjusted for time for follow-up from acute infection as a proxy for different variants of the virus.
The time frame of recruiting participants in the abstract (June 2020 to December 2022) does not align with that in the main text line 149 Page 5 (July 2020 to December 2022). Please clarify this.	Thanks for the comment, this is now changed to (July 2020 to December 2022)
Provide time points of measurements for lung function in the Pulmonary function section.	Thanks for the comment, this is now added.
Report p-values in Table 3	Thank you for this suggestion, the table is updated, and p-value is now added
	Thank you for these suggestions. We have now revised the text accordingly to the minor comments.

Minor comments:

1. Line 361 should be “upon further analysis, we identified four distinct clusters....”
2. Line 442, the sentence, “This cluster includes on daily function”, is repeated.
3. Remove the red underscores in Figure 2.
4. The legend title of Figure 3 should be "Cluster" instead of "clust"
5. Align the inner lines in Table 4

Dear Reviewers,

Thank you once again for your valuable and constructive comments, which have greatly contributed to improving our manuscript. We have carefully considered all feedback and made the necessary revisions, aiming to address each point clearly and thoroughly. We sincerely appreciate the time and effort you have invested in reviewing our work, and we hope that the changes we have implemented meet your expectations.

Best regards, corresponding author

Comment from the reviewers	Responses from the authors
Reviewer 2	
I applaud the authors of this work for how much they have improved the manuscript based on the reviewer comments. They have responded mostly satisfactorily to my previous comments and questions. Important references and explanations have been added, and (description of) statistical analyses have been improved. Listed below are my final comments to the authors of this manuscript.	Thank you very much, your comments have really helped us to improve the manuscript.
1. In my opinion the description of differences between self-referred and doctor/hospital referred participants in the manuscript is still lacking. Especially when claims such as “The NH-group reported significantly more symptoms compared to the H-group” are made in the plain language summary, it is essential to understand that hospitalized patients were followed up regardless of symptom persistence whereas nonhospitalized individuals were more likely to seek medical attention only if symptoms persisted, which the authors nicely pointed out in the response to reviewer comments.	Thanks for the valuable comment, we have now tried to clarify this in the method, discussion, abstract and in plain summary. -Methods, line: 167-171. -Discussion, line: 382-385. -In the abstract and plain language summary, we have removed the results of differences between the groups regarding symptoms.
2. I feel conflicted about the naming of the cluster groups as “recovery profiles”. Since this study was cross-sectional it did not study recovery of PCC, only the presence of persisting symptoms. No health status prior to COVID-19 or in the acute phase of disease was included with the exception	Thanks, we agree and has now changed this to cluster profiles instead.

of comorbidities. Referring to recovery profiles gives the perception that the study was longitudinal	
Reviewer 5 (response to R#3)	
The comments looks good to me, but I think the paragraphs about regression analyses need to be clearer:	
1. Please state the purpose of doing regression analysis, for example "To examine the factors associated with self-rated health" or something else. It is now not very clear what the analysis is for.	Thank you for a good suggestion, this is now added, line: 254-255.
2. Line 247 "All clinically relevant variables" is not clear, please list out the variables included in the model	Thanks for the comment, this has now been added, line: 255-262.
3. In the method section, it says that "Interaction terms between level of care (hospitalised vs. non-hospitalised) and selected covariates (dyspnoea, physical function, mental health, cognition, symptoms, and depression) were included in the main model" and "Based on the hypothesis that associations between clinical variables and EQ VAS may differ by follow-up time, level of care, or comorbidity burden, the model was extended to include relevant interaction terms." ". However, in the table of regression results (Table 3), no results for interaction terms was reported. Please either revise the methods or the result table to make them align with each other.	Thanks for pointing this out and we have now decided to remove this. In the analyses we also tested models including interaction terms (e.g., between level of care and selected clinical variables) to explore whether associations with EQ VAS differed across subgroups. However, adding these interactions did not significantly improve the model ($p = 0.097$). The explained variance increased only slightly (R^2 from 0.574 to 0.584), and the Bayesian Information Criterion (BIC) indicated that the simpler model was better. The inclusion of interactions did not change the main predictors, except for one MoCA interaction, which did not alter the overall interpretation of results. For clarity and ease of interpretation, and because the interactions did not improve the model, we now decided to present the results from the simpler model without interaction terms.
4. In the result section line 308 to 325, please signpost to the result table accordingly.	Thank you for the comments, this has been clarified in the text, line: 317-330.